# Assessment of lithium criticality in the global energy transition and addressing policy gaps in transportation

Peter Greim[1], A. A. Solomon [2✉] & Christian Breyer[2]

The forthcoming global energy transition requires a shift to new and renewable technologies, which increase the demand for related materials. This study investigates the long-term availability of lithium (Li) in the event of significant demand growth of rechargeable lithium-ion batteries for supplying the power and transport sectors with very-high shares of renewable energy. A comprehensive assessment that uses 18 scenarios, created by combining 8 demand related variations with 4 supply conditions, were performed. Here this study shows that Li is critical to achieve a sustainable energy transition. The achievement of a balanced Li supply and demand throughout this century depends on the presence of well-established recycling systems, achievement of vehicle-to-grid integration, and realisation of transportation services with lower Li intensity. As a result, it is very important to achieve a concerted global effort to enforce a mix of policy goals identified in this study.

---

[1] Institute for Material Resource Management, University of Augsburg, Universitätsstr. 1a, 86159 Augsburg, Germany. [2] School of Energy Systems, LUT University, Yliopistonkatu 34, 53850 Lappeenranta, Finland. ✉email: solomon.asfaw@lut.fi

Today's carbon-based energy system has negative impacts on environment, society and economy. In an age of population growth and rising energy demand, ongoing fossil fuel depletion and climate change call for alternative, sustainable solutions that depend on very-high shares of renewable energy (RE). Such a trailblazing change requires innovative concepts and technologies, including electrical energy storage systems for stationary grid applications in the power sector and mobile battery electric vehicles (BEV). Considering the outstanding dimension of quantities required for the global energy transition, questions of resource availability receive increasing attention[1,2]. Amongst others, one element comes to the fore—lithium (Li). Due to its function as a storage and flexibility option, a major technology application, the lithium-ion battery (LIB), takes on a fundamental role in fully RE systems as outlined in many studies[3–6].

LIBs achieved compound annual growth rates (CAGR) of 24%[7] from 2015 to 2018, driven by its increased use in power and transport applications[7,8]. Automotive applications constitute 70% of the total shipment in 2018, which was only 43% in 2015[7]. High energy density[9,10] and fast charging[11] will further promote this trend. The demand for automotive applications is estimated to grow by more than 30% per annum up to 2030[7]. Major battery manufacturers are committed to invest over 50 bUSD over the next 5 years to increase LIB production capacity, which is expected to exceed 1.2 TWh capacity by 2030[7]. Two key factors drive the increase in demand: first, the cost decline. During the last 5–10 years, tremendous reductions have resulted in the price of LIB packs falling to 300 USD/kWh in 2014[12] and 176 USD/kWh by the end of 2018[13]. Second, there is the influence of an experience curve. Cost improvements of $16 \pm 4\%$ per each doubling of historic cumulated capacity lead to 150 USD/kWh at 1 $TWh_{cap}$[14] for battery packs. In terms of time, fast learning might enable 124 USD/kWh by 2020[15].

Many studies have already tackled Li resources, their strategic availability and associated market policies[16–27]. Therein, several alarming questions are raised, but at present, the common understanding is that Li will not be a limiting constraint in this century. However, this consensus could be due to the scope of the studies to address the recent increased demand for batteries and various projections that show continued increase in its demand to support the achievement of very-high shares of RE-based energy supply[6,28]. Forecasts show that an increased demand for LIB will be due to a fast rise in BEV sales share[7,8,28–30]. For instance, these studies indicate that the steep part of the s-curve for the global EV sales projection starts around mid-2020, if the present market trend continues. Thus, to study the role of Li in the future energy system, an integrated and holistic demand assessment of all sectors is necessary. This requires the consideration of population and welfare growth as well as an associated significant increase in global total primary energy demand (TPED).

This paper investigates the long-term availability of Li by improving some of the detected limitations of past studies: key-detected weaknesses are: (i) EV growth projection that misses the latest market development; (ii) weak attention given to linking Li demand and the ongoing LIB driven effort to decarbonise the energy sector, among other things; (iii) the possible role of related factors, such as the secondary application of BEV batteries in stationary applications. A dynamic analysis of the supply and demand balance from now up to 2100 is performed in order to study how Li production (virgin as well as recycled) matches this new market demand. Eighteen scenario variations enable a comprehensive assessment of uncertainty and options. A century wide material flows could also show key issues of the coming decades in terms of use and production of Li. Finally, it will be shown that Li resource will be a cause of critical limitation for

long-term energy sustainability without any doubt, if clean energy transition is to be strictly enforced without ameliorating options.

## Result

**Li demand projection.** The concerns for Li availability are driven by expected demand growth associated with the significant increase in the LIB market. Hence, an understanding of application areas and their roles in major global demand trends is of greatest importance. In 2015, Li demand is about 34.6 kt. This spreads over EVs (14%), stationary ESSs (around 1%), traditional battery markets (25%) and non-battery applications (60%)[16]. A soaring demand for battery application over the last few years, with Li consumption share reaching more than 60%, was credited for the substantial increase in Li consumption to ~49 kt in 2019[31]. This trend is expected to continue in the coming years.

The estimate for stationary battery capacity for power applications up to 2050 is based on the results from LUT University energy system transition research[5,6]. The results estimate that battery capacity increases along with the growing penetration of RE to ~47.8 $TWh_{cap}$[6] in 2050. The second driving factor is the global increase of TPED. According to the United Nation's long-term target of global equity[32], a world population of 11.2 billion people[33] living at a European level of welfare will require 40 $MWh_{th}$ primary energy per capita[34] by the end of the century. Consequently, battery storage demand is scaled to 200 $TWh_{cap}$ by 2100 (Supplementary Fig. 1) because the total electricity demand by the year 2100 may be at least four times the electricity demand of the year 2050[34]. This gross estimate provides the basis of the potential upper limit of stationary LIB demand. Driven by cost advantages, mobile LIBs used in EVs are assumed to serve a second life (the remaining half life) as stationary batteries. Individual providers already claim its practicality[35,36]. Furthermore, other non-Li battery systems, e.g., vanadium redox flow or sodium–sulphur batteries, could share the stationary battery market.

In 2016, there are around one billion light duty vehicles (LDVs) on the road[37]. Following ICCT trends, this figure increases to 3.05 billion in 2050[37] (Supplementary Fig. 2). Thus, almost ten billion people[34] would have 0.3 LDVs/capita. This factor still exceeds today's global share of 0.16[33,38], but does not reach the European level of 0.48 LDVs/capita[38]. EV penetration, a term referring to the combined share of EVs (BEV plus plug-in hybrid electric vehicle (PHEV)) as a percentage of total LDV stock, is widely discussed in society. Due to its promise to become the least cost solution for transport modes[28], it is expected to enjoy an increasing market share though the projected rates of its increase varies depending on the source[13,28–30,39–44]. Despite major differences in the range of their projections, EV scenarios can be grouped into two. The first group assumes that EV sales shares continue to grow following a fast s-curve with proper policies, due to the dire need to decarbonise the transport sector[28–30,40,41,44]. For instance, these sources estimate global EV sales share to be 14%[28] by 2025; 40%[40], 48%[41] and 50%[30] by 2030; 100%[29] by 2050. These scenario projections suggest a faster rate of sales share increase even if one may be faster/slower than the other by <5 years. In addition to the specific projection discussed above, some of these studies present additional scenarios. The low scenario of Hummel et al.[28] expects 5.5% by 2025, suggesting a s-curve which lags by few years (~3 years). Such a delay has little impact on the result as presented below. The low scenarios of DNV GL leave several polluting cars on the roads by 2050[29]. The second group of scenarios associates the achievable EV shares strictly to the deployment of charging infrastructure, battery markets and ongoing regulatory policy changes, etc. Thus, they perform an in-depth analysis of what

needs to be achieved by various actors together with their EV projections. Typically, they have projections with lower market shares as compared to projections by the previous group. EV sales share of up to 33%[42] (IEA, 30@2030 scenario) by 2030, 57%[13] by 2040 and 66%[43] by 2050 was estimated. IEA also presents another scenario (named New Policies Scenario[42] (NPS)), which projects 15% global EV sales share by 2030. However, it should be noted that their NPS was created to analyse the sales growth based on existing policies and recent EV updates by ignoring commitments and potential improved policies. In this study, we adopted two EV sales share projections up to 2050 in order to closely evaluate the impact of the likely possibilities foreseen by both groups of forecasters. For the Best Policy Scenarios, we assumed 49% and 86% EV sales share by 2030 and 2050, respectively (Supplementary Table 1), in agreement with Khalili et al.[44]. Though this target is lower than the trends assumed in the other studies, such as DNV GL[29], it is considered to be a suitable target to achieve the required emission reduction in the transport sector to keep global temperature rise at ~1.5 °C by 2100 compared to the pre-industrial age. The corresponding numbers in terms of effective EV penetration, which is calculated as a BEV equivalent for the entire EV stock, are 18.4% and 79.8%, respectively (Supplementary Fig. 4 and Supplementary Table 1). Similarly, for the second scenario the share of newly sold vehicles is assumed to be 33% and 65% EVs by 2030 and 2050, respectively, in agreement with trends that the second group projects based on current EV initiatives and recent EV updates. Because of the encouraging trend in the industry and policy arena, we assume that EV growth may not be much lower than the scenarios of the second group. Thus, we excluded very pessimistic EV growth projections, for which sufficient data are also available[16–27]. Moreover, any lessons to be obtained by including additional low scenarios can be understood from the results presented in this paper. Because the various studies referred to above also assume different LDV stocks[13,28–30,39–44] by 2050, this study also includes 2 billion final LDV stock by 2050 for a low demand scenario, while applying both EV shares increases assumptions. Note that LDV stock was assumed to remain constant for all years after 2050 for both the cases, though intuition suggests an increase. The remaining share of the 100% renewable LDV transport is assumed to be provided by alternative concepts like power-to-liquid, biofuels, power-to-gas, EV with new battery chemistry or fuel cell electric vehicles based on hydrogen. However, it could be noted that extreme scenarios may still rely on some fossil ICEs by 2050, particularly for the trend that follows the second EV projection. The EV market is assumed to be constituted by BEV and PHEV. The assumed average battery capacity per vehicle is set to 60 kWh for BEV and 15 kWh for PHEV, which will have 8 years of lifetime serving in EVs[16,28,45]. At the end of their life, these batteries will be used for an additional 8 years as stationary batteries before entering a recycling loop. The assumed 16-year service time is lower than the 20-year lifetime that the industry provides for stationary batteries in agreement with Turcheniuk et al.[46].

Though recycling LIB is still under development, reports show that ~97,000 tonnes of LIBs were recycled globally in 2018[47]. The slow development is due to economic reasons[48] and lack of regulations, as well as challenging technical processes and collection procedures. The Li recycling efficiency was set to 95% based on recent technological development[47–53], while the collection rate was set to grow from ~45% at present based on global data[47] to 99% by 2050 (see 'Methods' and Supplementary Fig. 5).

Whenever efficiency improvements of LIBs fit to the requirement of city and intercity buses, electric bikes and scooters as well as medium duty vehicles (delivery trucks) and heavy trucks will use batteries. A growth curve is applied to provide the projection for these types of applications, which are assumed to require 50 TWh (average) battery capacity in 2100 (Supplementary Fig. 6).

To convert the battery capacity to the equivalent Li requirement, a long-term estimate of Li intensity per storage capacity of ~130 g/kWh$_{cap}$[16] is applied uniformly up to 2100, which is at the bottom range of literature data (Supplementary Table 2). However, future research should employ insights from presently missing Li intensity learning curve when such data are available.

In addition to these quantitatively dominating applications, Li is also used in varying industrial applications, as batteries and for non-battery use. Based on current annual demand[16], the corresponding Li demand is calculated with a CAGR of 3% and 2%, respectively, following a recent global economic growth trend[54] (Supplementary Figs. 7 and 8).

This study creates eight demand variations (Supplementary Fig. 9) by combining relevant factors, which are used to create 18 scenarios together with 4 supply scenarios to be discussed in the following section. While addressing the low demand cases, our scenario definitions lean towards investigating the possible challenges of the aspired transition to sustainable energy systems, for which at present EVs are the best candidate to meet the climate change mitigation targets in the transport sector.

**Li supply from resources to production output**. In 2016, global Li supply is 38 kt[31] (Supplementary Table 3). Because of its very-high chemical reactivity, Li has no elemental occurrence in nature, but can be mainly found in ionic compounds like oxides or chlorides[26]. These are enriched either in ores as minerals (Supplementary Fig. 10) or in salt solutions as brines. Both major types of deposits differ in geological formation, extraction and process technology, associated costs and time, sustainability as well as size and dispersion. Furthermore, Li is dissolved in oceans as an almost 'unlimited' resource. Due to poor maturity of the extraction techniques and expensive production costs, seawater extraction is not expected in the near future[20,26].

To focus on strategic and long-term aspects, this study is limited to examining resources that are geologically confirmed without considering the restrictions concerning socio-economical exploitation or current state of technology. The latest data from the United States Geological Survey (USGS) indicate total resources of 80 Mt Li[31]. However, an in-depth literature review reveals the subjective, non-transparent and imprecisely defined character of resource estimation. Figures ranging from 30[20] to 95 Mt Li[26] differ by more than a factor of three (Supplementary Table 4). Due to these divergences, this study uses four scenarios covering one low (26 Mt Li), one medium (41 Mt), one high (56 Mt) and one very-high (73 Mt) resource value (see 'Methods' and Supplementary Table 5). The lowest number covers the range of proven reserves[26,55] and describes a worst-case situation, where no additional resources are exploited. Both next higher assessments, in turn, assume the potential extractable mineral deposits. Notice that, as shown in Fig. 1, all deposit costs are lower than the price of industrial grade Li$_2$CO$_3$, suggesting their economic viability depends on time. The 41 Mt reserve estimate is based on the higher range of the proven mineral reserves, which is below the red line, and as shown by the yellow line. The value of 56 Mt corresponds to the more optimistic reserve quantity, which assumes that all reserves could provide their estimated high resource potential. The very-high reserve covers the range of some very high, but due to missing rationale, rather unrealistic estimates[25–27].

Geographically, Li deposits are distributed rather unequally on a global scale (Supplementary Fig. 11). Effects on social and political interests as well as economic trading are important[17].

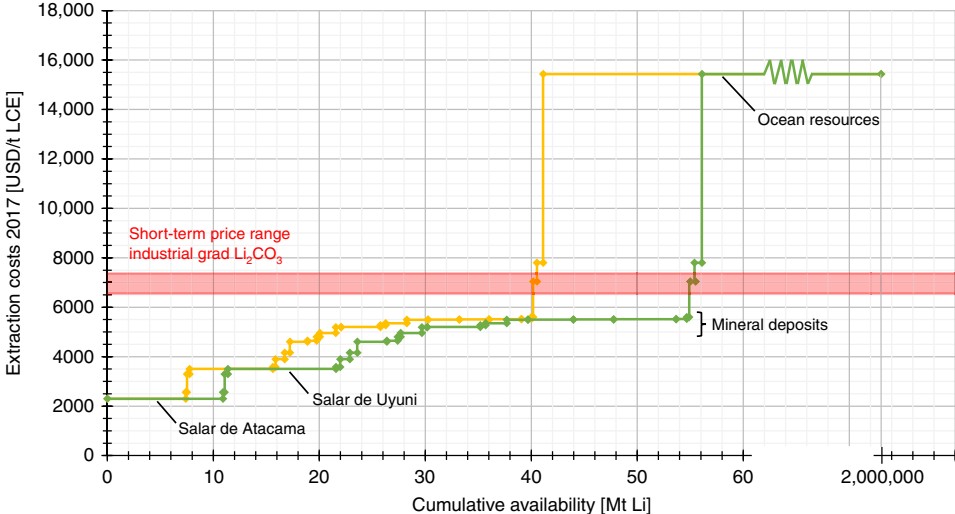

**Fig. 1 Cumulative availability curve of global Li resources.** The availability curves of this study's medium (yellow line) and high resource values (green line) are plotted. Both reveal four major plateaus. Beginning with cost leading Salar de Atacama, South America's second major brine, Salar de Uyuni, accounts for the next horizontal line. The majority of mineral deposits follow at some distance. The date at which society must switch to these more expensive deposits depends on the amount of available resources. This equally applies to the shaping leap to ocean resources, the height of which, in turn, depends on the extraction costs of Li from seawater. Current estimates suggest different values framing a range of 3[20, 27]−30[26] times the usual costs of pegmatites and brines (Supplementary Table 6). The red coloured area marks the expected short-term price range for industrial grade lithium carbonate. Nearly all conventional deposits are below. As a consequence, the values for presumed medium and high resource assumptions are economically justified. Note that the extraction cost estimate on y-axis is per lithium bicarbonate equivalent (LCE).

Because to bring these resources out of the ground, exploitation must pay off; the marketable price must exceed extraction costs. For a long-term assessment of the latter, the concept of the cumulative availability curve[56] is used (see 'Methods' and Fig. 1), which specifies the amount of resources being available at certain costs. In theory, society must extract the next more expensive deposits as the demand for the resource increases. Because of a less energy-intensive extraction process, brine deposits are generally cheaper. Their extent, particularly in large South American 'Salars', determines how long these low-cost resources are available. But respective to the current lithium carbonate ($Li_2CO_3$) prices[57] and the continuing demand, increased extraction costs may not restrict the availability of Li.

However, time has its own constraints. In the build-up phase, so-called greenfield projects must go through resource discovery, several stages of feasibility studies, facility construction and production start-up. This usually takes one to two decades[16,20]. After, the process time along the value chain determines the flow rate of fresh material into society. The lead time of $Li_2CO_3$ appears to be uncritical for mineral deposits (e.g., 5 days for spodumene treatment[17]) but becomes a limiting factor for brines. Relying on solar irradiation, the evaporation process is not constant throughout the year and takes 1–2 years[17]. Even perfect conditions as found at Salar de Atacama delay the production for at least 12 months[20]. The Li supply system implies a certain moment of inertia.

To quantitatively assess Li supply, the inflow of virgin material expressed by the production volume per year is modelled by applying logistic-growth-based bell-shaped curves following Vikström et al.[26] (see 'Methods' and Supplementary Fig. 11). The bell-shaped curve is fit to the historical production data to choose the curve that produces short-term projections in agreement with recent developments. However, because fitting to the present production trends leads to a sharp rise in supply as compared to the demand projections, we enforce a criterion that production around mid-century is not larger than 10% of the annual base case demand. Yet for the high and very-high

production scenario, some years see an over production as high as 30% for the same demand due to the demand curve that has a shape of a roller coaster. (Supplementary Fig. 9). As opposed to the production estimated by Vikström et al.[26], this study is much more comprehensive because of included resource scenarios.

**Critical dynamics of Li**. As a first step, we present the comparison of supply and demand (see 'Methods' and Fig. 2) on a yearly basis. Figure 3 reveals highly critical dynamics using various scenarios. Figure 3a reveals that Li production shows a good balance with annual fresh Li demand in the near-term for the medium production for almost all policy scenarios. However, Li supply and demand balance start to show strong demand scenario dependence around 2030. Focusing on scenarios related to EV shares, it can be seen that for the Best Policy Scenarios (BPS 3b LDV) the observed good balance of Li demand and supply extends to ~2050, when it reaches a time the market started to experience a large deficit that lasts for the remaining half of the century. The inflow of virgin material and the increase in recycling is not sufficient to supply the important transition years for most part of the second half of the century. The supply and demand balance was found to show large surplus from 2030 up to 2050 for other EV-related Li demand as can be seen from the curves corresponding to the three scenarios, namely BPS 2b LDV LD, CPS 3b LDV and CPS 2b LDV. In particular, the CPS 2b LDV scenario resulted in larger surplus, which extended up to 2053 when the deficit dominating the remaining part of the century begins. The appearance of such an early deficit following two decades of large surplus is obviously not a good representation of the real market operation for these scenarios. In reality, while a short-lived large surplus may be possible, the long-term market should show stable demand and supply balance, except for the case of not manageable deficits caused by resource scarcity. The reported special surplus occurred because the production was modelled solely focusing on BPS 3b LDV demand in

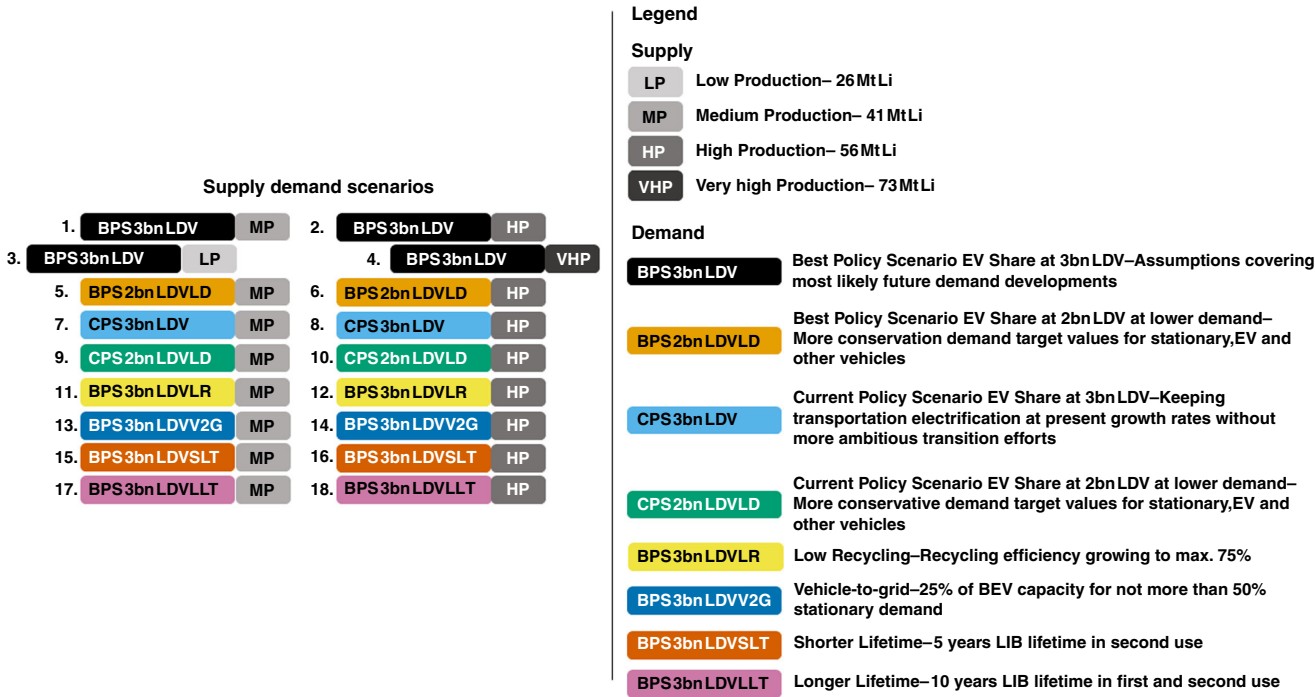

**Fig. 2 Scenario table.** Listing and description of this study's supply and demand scenarios. In total, there are 18 pairs that consist of one demand element and one comparative supply element each. Two Best Policy Scenarios (1., 2.) are the groundwork for 16 subsequent deviations, which cover the five most important influencing factors (supply, EV uptake policy, recycling, V2G and battery lifetime). The high supply uncertainty requires two base supply scenarios. The legend describes the individual components of this modular system.

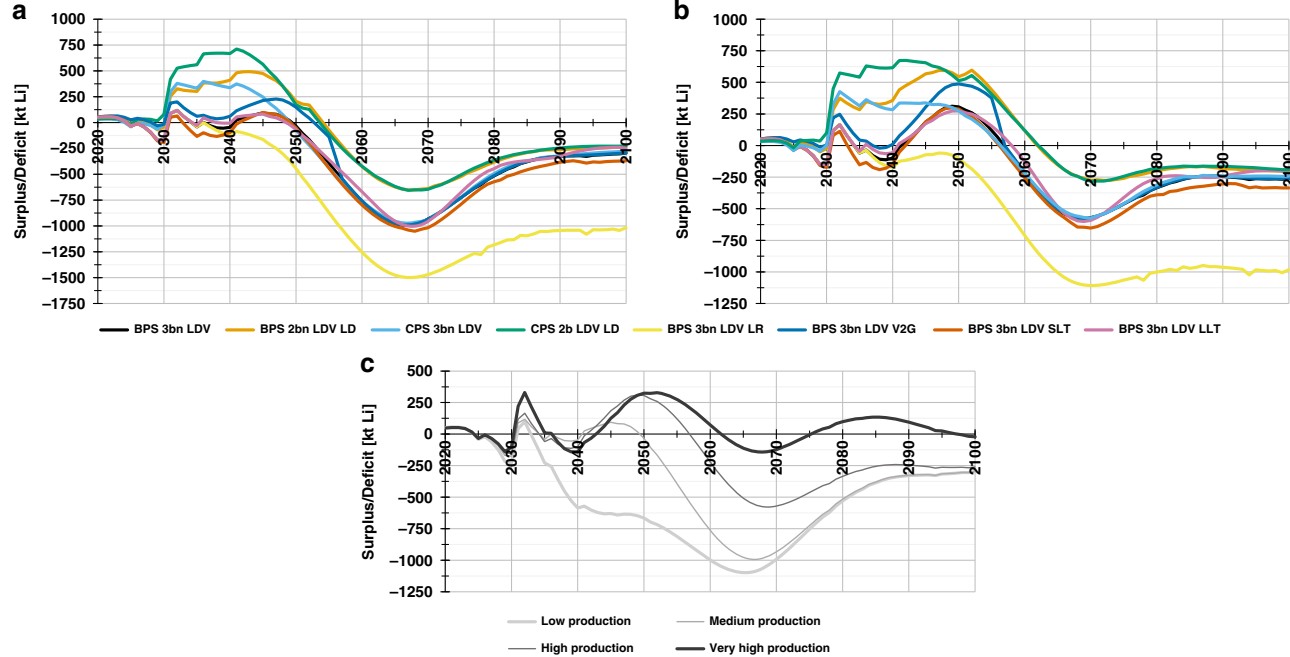

**Fig. 3 Annual comparison of Li production and fresh demand.** Annual Lithium supply and demand balance. The annual surplus or deficit of lithium for **a** scenarios involving medium production; **b** scenarios involving high production; **c** various production scenarios under the BPS 3b LDV demand scenario.

order to simplify inter-scenario comparison rather than producing production scenarios that fit to each demand curve. This is what led to an extended large surplus for lower demand scenarios as observed in these cases. To address this gap, we assumed that the surplus will be accumulated for later use (Supplementary Figs. 13–15). In this circumstance, the year the deficit occurs moves to a later year, though the amount of the shift depends on the demand scenario. For these three scenarios, the shift is larger. Specifically, the deficit moves from 2054 to 2077 for the CPS 2b LDV demand scenario. Three of the remaining four curves, which correspond to battery lifetime in second use (represented by the remaining two curves presenting variation in lifetime of secondary batteries), and vehicle-to-grid (V2G) integration, closely follow the trend corresponding to the BPS 3b LDV demand with

some minor differences depending on the demand scenario. For instance, the integration of V2G technology[40,58] relieves the power sector. Hence, less second-life material is required and more Li can go directly to recycling because of BEV integration into the grid. This retention of Li in the loop pulls the corresponding graph a little up. The last remaining scenario, namely the BPS 3b LDV LR demand, was an exception in generating deficits well before 2040 that got worse for the remaining part of the century as shown by the drastic deficit. This is strong evidence of the dire need for the establishment of an efficient Li recycling system. Under the high production conditions given in Fig. 3b, all scenarios maintain similar trends to the corresponding scenarios discussed in Fig. 3a, but the additional supply pulls up the curves with a little delay of the start of the deficit in the second half of the century. Accumulating the surpluses also shifted the time when this deficit starts for all scenarios as shown in Supplementary Fig. 14. Specially, the CPS 2b LDV experiences no deficit throughout the century for this supply. Such a significant change also shows that the deficits are mainly due to transport sector Li demand.

The foregoing result shows that the balance between supply and demand depends on the presence of well-established recycling systems, Li production rates, achievement of V2G integration and the ability to limit LDV stock growth without compromising the transportation services that society needs to function efficiently. This finding has important lessons regarding future policy directions that should be pursued in order to achieve a sustainable transport sector that can conform to the sector's emission target of keeping temperature rise at about 1.5 °C above pre-industrial levels by 2100. The above result clearly shows that scenarios that can conform to the stated climate target and improved transport equity will definitely result in serious Li supply deficits over the next century. On the contrary, low demand scenarios, such as the CPS 2b LDV or lower, achieve a balanced Li supply and demand throughout the century. However, studies show that such scenarios definitely compromise the climate change target[59]. To solve this conflict, global concerted effort is required to acquire commitment to enforce the mix of the following policies across the globe: (i) develop transport services that could reduce the dependence on LIB (reduce number of LDV) by promoting improved public transportation, shared rides and other possible solutions; (ii) establish and maintain efficient recycling system; (iii) improve LIB technology to reduce material demand per battery capacity; (iv) develop new battery chemistries or other sustainable transportation options that will reduce the demand for LIB.

In addition, we present the impact of deviations of production as given in Fig. 3c. The effect of different supply estimates is not visible until 2028 because all production projections are fitted to historical production. After 2030, however, the curves start to diverge. At low resources, a continuous deficit runs through the entire century. The aforementioned effects of TPED increase reflect a clear dip. Very-high resources, in turn, enable partly significant surpluses around 2050. For this scenario, the observed deficit can be covered by accumulating these surpluses to achieve the supply–demand balance throughout the century.

**Material flow of Li during this century**. In a second step, the availability of Li is examined at a century level in order to clarify the capability of the estimated resource potentials to cover the demand dynamics over longer time periods. Figure 4 reveals one rather clear message: the penetration of storage systems based on LIBs results in a prospective availability constraint of Li during this century. Nearly all considered scenarios run into a—varyingly strong—deficit. However, at high supply condition, demand

of the CPS 2b LDV and BPS 2b LDV scenarios could be matched by the available resource throughout the century.

The very-high resource condition was the only case that can match the BPS 3b demand scenario (one of the highest demand) and thus can meet all other demand scenarios throughout the century. The depletion year given on top of each bar in Fig. 4 also shows that the fewer the resources, the earlier they are depleted and the higher the resulting deficit.

Now, let us examine the material flow using the BPS 3b LDV scenario to understand the process. The deficits in the BPS 3b LDV scenarios (except in the very-high supply condition) are a consequence of the cumulative base case demand of 68.03 Mt of fresh Li until the year 2100. The way of this material flow within the system can be traced in Fig. 5, which presents a hypothetical scenario by assuming that the required fresh Li is available (a condition that is possible for the very-high resource scenario only). This chart is of greatest importance as major correlations of application areas and energy sectors become clear. The bulk of inflowing fresh material is used for BEVs and—a little less—other transport applications. This huge stock continues in a flow towards second-life material and feeds almost the entire stationary sector. Only a tiny amount of net demand remains to be supplied by fresh material. All spent batteries go to recycling —irrespective of whether directly after first use or after second use. The rest of the material, which is currently in use, forms the fictional volume of Li in stock. Applying the base case demand, this amounts to 51.29 Mt in the reporting year 2100. The difference of 16.74 Mt leaves the system and is lost. This drain consists of losses due to collection rate and recycling efficiency less than unity as well as all industrially used material, which is not recovered.

In consideration of the significance of annual dynamics and the size of the cumulative recycling loop, the return flow of secondary material back to society is one fundamental part of the model. A well-established and highly efficient recovery system is essential to maintain Li in circulation to augment the supply shortage. The consequences of a less efficient recycling system can be seen from the corresponding deviation that shows vast deficits at the end of this century. At medium resources, Li deposits are already depleted in 2055.

## Discussion
In this article, the availability of Li is assessed in consideration of the forthcoming energy transition towards very-high shares of RE supply. The analysis comes with two major findings that clearly show the criticality of Li in the form of long-term supply shortage.

First, the expected demand growth could be matched by the projected production scale-up for almost all scenarios, over the next two decades. The short-lived, scenario-dependent, small deficit between now and 2050 can be managed with minor adjustment for almost all scenarios, except for the scenario corresponding to low recycling for which an early supply deficit, that continued until the end of the century, occurs due to an early depletion of fresh Li supply. Thus, maintaining the good balance of supply and demand in the first half of the century requires the development of an efficient recycling system. Even if that is done, maintaining good balance up to the end of the century is only possible, if the Li resource availability is at least as high as the very-high-production scenario estimated in this research or, if the number of LDV is limited to two billion and the corresponding Li production is at least similar to the high production scenario. Despite a clear evidence that lower EV uptake eases pressure for Li demand, for the present assumptions its impact is not as strong as maintaining lower growth in LDV population. But note that

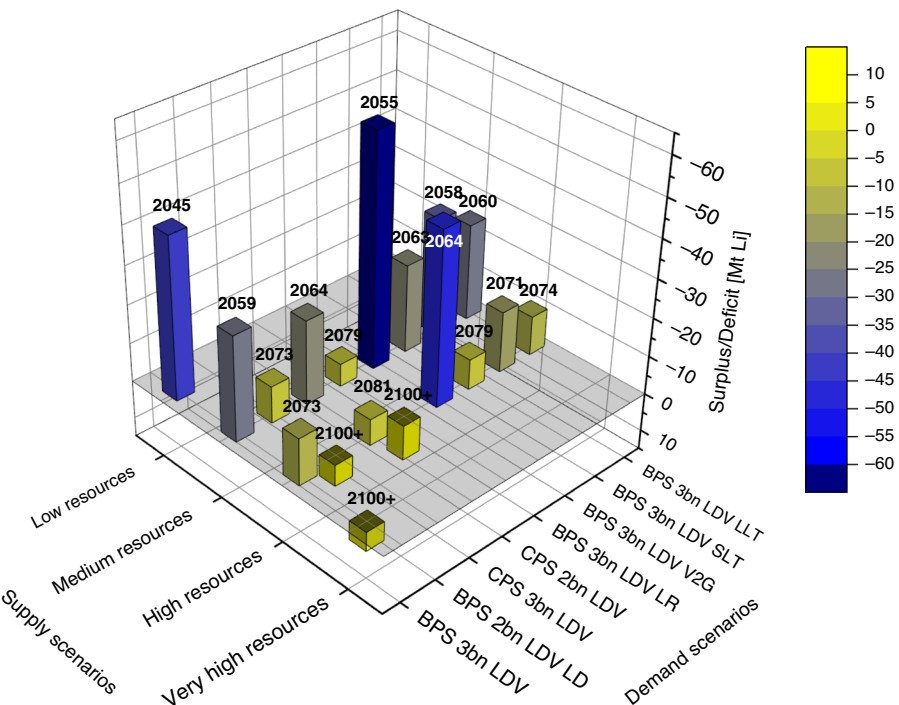

**Fig. 4 Availability of lithium in the year 2100.** The comparison of resources and demand represented by cumulative drain plus Li in stock is shown in 18 scenarios of surplus and deficit, respectively. Above each column, the year of depletion is indicated by case.

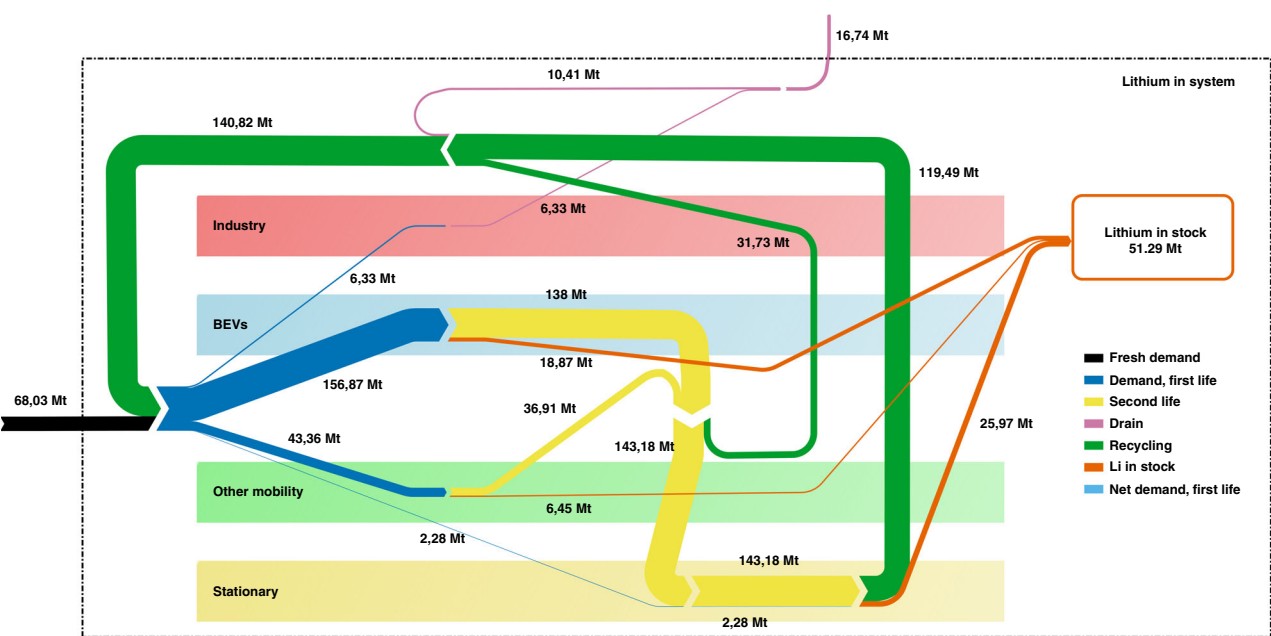

**Fig. 5 Lithium flow until the year 2100.** The material flow analysis visualises the sectoral interaction of the integrated demand projection model. The total figures of the base case demand are used in this case. Accordingly, 68.03 Mt of inflowing fresh Li splits into four streams supplying—together with backflowing recycled material—the demand of the considered fields of application. The gross demand of BEVs is quantitatively standing out. A strong flow of second-life material is used in the stationary sector, reducing its net demand to 2.28 Mt Li. Except for the material currently used in the year 2100 stock, a strong recycling loop maintains the lion's share of Li in the system. Only 16.74 Mt of Li flows out through industrial applications that are not recycled at all and losses within the recycling process.

limiting EV uptake growth to a rate lower than the CPS 2b LDV scenario could have as much or even larger effect in easing Li demand. However, studies show that low rates of EV uptake will compromise climate change targets by favouring massive use of fossil-based ICE vehicles and related emissions.

Second, in agreement with the annual dynamics, in the century level cumulative analysis, existing resources supply the demand throughout the century only in few cases. Even for the scenarios with century level supply balance, the supply shortage could appear if the analysis is pushed by some years beyond 2100.

Interestingly, those scenarios correspond to the very-high resource and the high resource scenario with lower demands. For all other cases, supply deficits result in accompanying resource depletion before end of the century. Contrary to other assessments[20,27], the result shows that Li availability will become a serious threat to the long-term sustainability of the transport sector unless a mix of measures is taken to ameliorate the challenge. The mix of these ameliorating measures are: (i) reduce the dependence on LDV and thus LIB by promoting improved public transportation, shared rides and other possible solutions; (ii) recycle once produced LIB by establishing and maintaining efficient recycling systems; (iii) improve LIB technology to reduce material demand per battery capacity; (iv) substitute demand for LIB by developing new battery chemistries; (v) replace demand for batteries by developing sustainable transportation options that do not require batteries. Key reason for the difference between our results and previous studies is the assumed low battery demand in those studies, which misses latest EV uptakes, respective cost progress, upcoming potential and massive pressure to reach ambitious climate targets, as documented by the European Green Deal[60]. Note that the major driver for the observed deficit is the significant use of LIB in the transport sector. Besides the transport sector, pursuant to the effects of population and welfare growth, the resulting increase of TPED keeps the demand at a high level. Nevertheless, if managed well during the second part of this century, there is enough time for mitigation strategies to take effect. High-performance battery concepts[61,62], like Li-air or Li–S, as well as additional resources out of oceans are most promising. Equally, the development of substitution technologies like batteries based on aluminium, sodium or magnesium should be encouraged[63–65]. ICE vehicles could be also fuelled with synthetic fuels, but for the price of low systemic efficiency and high economic cost[44]. Substituting Li ion as a stationary battery may be possible in near future but its effect on Li availability may be minor over the long term because the observed fresh Li demand is mainly driven by electric vehicle adoption. This is due to the availability of large second-life batteries that can cover the need for new utility-scale battery, its substitution has led to insignificant change in the observed dynamics. Due to high performance requirement of transport applications, developing a chemistry that will compete with the present performance of LIB will take time. As a result, lack of data makes estimating the impact of such substitution difficult. However, the dynamic for the two EV uptake rates considered in this study is a good indicator that the alternative chemistry, even if developed, may not play a significant role in the next three decades because of the corresponding lead time. At the same time, because the advanced chemistries are still on a research level, it is not possible to evaluate the impact of replacing the present batteries with technologies that have lower Li intensity. However, although the natural resources might be exhausted, the significant amount of Li stock within the system can be recycled to create additional supply. Thus, maximum effort is needed to implement a highly efficient recycling process as fast as possible. Without such effort, material drain will eliminate the significance of LIB in the near future. In this regard, sector coupling via second-life usage is another 'must' requirement for policy making. Finally, the foregoing result clearly shows that Li supply is very critical to energy transition. But the level of its criticality depends on demand scenario and the corresponding Li reserve. Even moving to the high Li resource scenario does not eliminate its limiting effect except for the unlikely very-high resource scenario. Similarly, reducing demand did not fully eliminate the challenge but added the risk of compromising climate change goals. However, caution will be necessary because of the inherent multiple sources of uncertainty of such studies.

In summary, present production trend shows that in the short term, supply and demand is well balanced but the long-term sustainability of the transport sector is at risk. At present, a concern on climate actions dominates discussions; however, it is equally important to address policy gaps in order to address the embedded long-term risk of sustainable transport sector pathways. To address these gaps, a concerted global effort is necessary to enforce well-established recycling systems across the globe, enhance transportation services and battery performance to achieve a lower Li intensity in the sector, while also improving efforts to develop alternative options. The significance of the obtained results calls for further actions and investigations in this area.

## Methods

**Supply scenarios**. This study's resource data rely on the figures of Vikström et al.[26]. The low and the high values refer to indicated minimum and maximum values, respectively. To eliminate the speculative component, only deposits with information about the Li content are considered. The upper value of Salar de Atacama is set to 10-Mt Li, covering a more justified range[17,55]. The medium value is the mean value of low and high estimates. The very high is taken from Sverdrup[25], a value that is 7 Mt lower than the latest estimate of the USGS[31].

**Cumulative availability curve**. The available resource, which is represented by the cumulative availability curve, depends on the commodity price of Li. Because of the difficulty to estimate the commodity price, extraction cost is used to show its effect on the cumulative availability curve. For the extraction costs of the $y$-axes, currently known information[66] serves as starting point. To do justice to anticipated long-term efficiency improvements, the figures are reduced by 200 USD each. The rest of non-public and not yet calculated cost data (USD*$t^{-1}$) is determined by a common correlation analysis using the specifications on Li content (%) and Li concentration (%)[26], respectively. The plots (Supplementary Figs. 16 and 17) result in regression formulas for brine (Eq. 1) and mineral deposits (Eq. 2).

$$\text{Extraction costs (USD*}t^{-1}) = -19,656 \cdot \text{Li concentration} + 5391.3, \quad (1)$$

$$\text{Extraction costs (USD*}t^{-1}) = 73.526 \cdot \text{Li content} + 5464.6. \quad (2)$$

The list of all extraction costs can be seen in Supplementary Table 7. For the sake of clarity, mineral deposits below 0.5-Mt Li are rounded to full hundred numbers building various collection indices.

The lower cost limit of Yaksic and Tilton[27] and the theoretical resource value of Vikström et al.[26] build the quantitative framework of ocean resources (Supplementary Table 6).

**Production projection model**. The exploitation of resources follows a logistic relation. Its most general form is described by the lower (parameter $A$) and upper asymptote ($K$), the scaling parameter ($Q$), the growth rate ($B$), the time of maximum growth ($M$), and a parameter affecting near which asymptote maximum growth occurs ($v$)[33].

$$f(t) = A + \frac{K - A}{(1 + Qe^{-B(t-M)})^{\frac{1}{v}}}. \quad (3)$$

To address market dynamics throughout the whole century, this study uses two different sets of parameters for the years until 2030 and beyond. In the first section, the curve follows historical and near-term predicted production data. The parameters of the applied logistic curve can be seen in Supplementary Table 8. The scaling parameters $v$ and $Q$ are fixed at 1 to keep the model easily treatable and modifiable. $K$ describes the amount of available resources in kt Li. The growth $B$ is adapted to historical production[16,55]. The peak year of production, equivalent to the time of the logistic curve's maximum growth M, follows subsequent regression function

$$\text{Peak year} = 11.298 \cdot \ln(\text{resource base}) + 1941.3. \quad (4)$$

To determine this, varying resources and resulting peak years have been plotted as a fixed growth parameter (Supplementary Fig. 18).

In the second step, the curve is adjusted to balance our demand model as good as possible after 2030. The parameters are chosen individually to respect market's compensation function and to prevent unrealistic high surpluses or deficits. The parameters of the applied logistic curve can be seen in Supplementary Table 8. The respective production rates are the annual change in cumulative resource values, in both cases.

**Demand projection**. The operating principle of the integrated demand projection model coupling mobile and stationary applications is explained by means of the sample BPS 3b LDV (Supplementary Figs. 19 and 20).

The calculation of required BEVs is based on two logistic curves. The first refers to LDV stock growth. Associated parameters can be seen in Supplementary Table 8. Its derivate describes the amount of additional sales required to realise this increase. The sum of this figure and all scraped vehicles gives the amount of annual new LDV sales. BEVs are scrapped after a lifetime of 16 years and other LDVs after 12–14 years[67,68]. The amount of appropriate new BEV sales is obtained by multiplying by the respective percentage share in sales following the second logistic-growth assumption. Its parameters can be seen in Supplementary Table 8. All BEVs are replaced with one fresh LIB after 8 years, or half of the lifetime. Hence, the demand of Li doubles in a term of BEV equivalents, which is the sum of new BEV sales and replacements. Once used, LIBs begin their second life in stationary power applications. If not required, they directly enter the recycling loop.

Due to similar characteristics and performance requirements as well as a common environment, other transport applications are treated equally. Specific stock growing parameters can be seen in Supplementary Table 8.

Inspired by comparable market structures, stationary applications follow the anticipated penetration of RE for the period after 2050. Associated parameters can be seen in Supplementary Table 8. Again, the total demand of LIBs consists of stock growth generating sales plus the reconstruction of scraped batteries. First life stationary LIBs get scrapped after 16 years; second-life LIBs after 8 years. Resulting gross demand is reduced by the amount of second-life batteries coming from BEVs and other transport use. The entire scrap goes into recycling.

The recycled material goes back to circulation in the year where the end-of-life state is reached. Hence, Li that is not used for second life creates additional supply one lifetime earlier.

In contrast to former logistic functions, the calculation basis of industrial applications, CAGR follow a declining curve given in Eq. (3). Decline parameters can be seen in Supplementary Table 8. Due to a missing economic incentive, problematic technical feasibility of non-battery use and tricky collection of privately used portable LIBs, no recycling for industrial applications is assumed.

**Supply and demand analysis**. The supply and demand projection models output gravimetric figures on an annual basis. The respective comparison is the result of the trivial mathematical operation of a difference. The century wide cumulative perspective, however, is more comprehensive. Whereas supplying resources are fixed to one initial value, the required material steadily increases with time. To be independent of one specific reporting year, it was deemed appropriate to illustrate the demand side by means of the quantity of 'Li in stock plus drain'. The stock consists of the respective material that is used in BEVs as well as stationary and other transport applications. Industrial applications are assumed to be used within 1 year and do not contribute to the stock. The second cumulative value, Li drain, is nourished by aforementioned industrial applications and losses of collection rate and recycling efficiency less than unity. By assessing this value, the amount of material that leaves the system and is irreversibly lost is determined. This flow quantity might be most important for future issues of Li availability.

**Scenario variation**. The database and some additional information on selection and significance of this study's demand scenarios are summarised as follows.

Due to the high divergence of resource estimates, an examination of lower and upper limits is indispensable to frame the range of conceivable developments. Hence, the scenario 'Supply' (Supplementary Figs. 21 and 22) looks at low and very-high resource availability. Both figures are compared at the BPS 3b LDV demand.

The other scenarios refer to 'Demand' modifications. They are compared at medium and high resources each. Not-mentioned assumptions remain unchanged from the BPS 3b LDV demand.

Due to its highest influence the EV penetration steers the first scenarios. The first deviation refers to the 2050 target LDV fleet that is only 2 billion vehicles (Supplementary Fig. 2 and Supplementary Table 8). In addition, the BPS 2bn LDV LD scenario (Supplementary Figs. 23 and 24) deals with lower demand in a situation where the deployment of LIBs is reduced. Less comprehensive TPED increase or the rise of competitive technologies are possible reasons. Hence, stationary and other transport applications are curtailed by 25% each reaching 150 and 37.5 TWh in 2100, respectively (Supplementary Figs. 1, 6 and Supplementary Table 8). For stationary applications, battery systems not depending on Li reduce the demand of LIBs. Considering eligible batteries' early stage of maturity, the contributing share increases from almost zero to 50% by 2050 (Supplementary Fig. 25 and Supplementary Table 8).

Considering lower EV sales and growth rates the Current Policy Scenario looks at the situation for 3bn (Supplementary Figs. 26 and 27) in the CPS 3bn LDV scenario and 2bn (Supplementary Figs. 28 and 29) in the CPS 2bn LDV LD scenario. The second one also covers the above-mentioned lower demand assumptions in stationary and other transport modes.

The variation low recycling (Supplementary Figs. 30 and 31 for the BPS 3bn LDV LR scenario) deals with a situation where the target recycling efficiency is missed. As large-volume implementation of theoretical research results is not yet practically proved, less efficient processes and their effect on availability must be investigated. Hence, a value of 75% is used (Supplementary Fig. 5 and Supplementary Table 8). The collection rate, however, is maintained.

The scenario 'V2G' (Supplementary Figs. 32 and 33) for the BPS 3bn LDV V2G scenario respects an up to date technology, for which sector coupling economic benefits accurately meet this study's holistic approach. However, due to grid requirements the contribution is limited, since about two third of all batteries are most likely not at the distribution grid level, where V2G is applied[6]. Numerically, the use of 25% of BEV capacity for not more than 50% of stationary demand is assumed.

The variations shorter (Supplementary Figs. 34 and 35) for the BPS 3bn LDV SLT scenario and longer lifetime (Supplementary Figs. 36 and 37) for the BPS 3bn LDV LLT scenario deal with the situation of modified LIB duration. In the former, fast performance degradation is considered by decreasing the second lifetime to 5 years. Anticipated long-term technology improvements drive the assumptions of extended lifetime. Accordingly, 10 years for first and second life each result in a lifespan of 20 years in total.

**Recycling**. The recycling data starts at current figures[69] and takes fast learning as a basis to end up at indicated final values by 2030. Growth parameters can be seen in Supplementary Table 8.

Due to our approach of using LIBs in a second life, this model's recycling data differs from industries' mid-term expectations[46]. Removing this factor our data is directly comparable (Supplementary Table 9).

**Reporting summary**. Further information on research design is available in the Nature Research Reporting Summary linked to this article.

## Data availability
The authors declare that the data supporting the findings of this study are available with in the paper and its Supplementary information files.

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

## Acknowledgements

The authors gratefully acknowledge the support from the LUT internal research platform REFLEX. A.A.S. would like to acknowledge partial funding from Academy of Finland, for investigating biophysical limits of energy transition (317681). The authors would like to thank Michael Child for proofreading and Lars Wietschel for supporting figure preparation.

## Author contributions

All authors jointly designed the study, created the database and developed the model. P.G. implemented the findings and wrote the paper. A.A.S. supported the structuring and writing. C.B. supervised the research, guided the study and edited the paper.

## Competing interests

The authors declare no competing interests.
