## [Peer Review File · Nature Communications]

REVIEWERS' COMMENTS:

Reviewer #1 (Remarks to the Author):

Overall this is an excellent paper. While it covers previously studied ground using very similar methods it does bring fresh assumptions and contemporary data, giving the analysis merit. The paper is also comprehensive, with broad recognition of the existing literature and sufficient detail regarding assumptions and methods. A few specific points below:

- There is a reference missing in the second paragraph
- the second bullet in paragraph 4 page 1 does not make sense to me. Does this mean that previous studies have not done enough to understand other, non-automotive uses of lithium ion batteries in the energy system to meet decarbonisation targets? clarify please
- The certainty with which the closing statement of paragraph 4 concludes the introduction belies the various uncertainties discussed in the paper, the widely acknowledged challenges of forecasting natural resource production, and the inherent uncertainty in forecasting in general. Particularly given the papers conclusions that most scenarios do not see critical limits to lithium supply in the next two decades. 20 years is sufficient time for a number of changes on the supply or demand side to impact the chance of critical limits to the lithium supply/demand system
- The ameliorating options mentioned in the paper could be laid out more clearly in the papers conclusions. A bullet list of the things that your analysis suggests would be most useful would be sufficient
- It is more accurate to refer to the Cumulative Availability Curve as presenting the available resources given a particularly commodity price, not production cost.
- The paper closes with a statement that lower lithium intensity in batteries could help ease longer term constraints. However, the paper appears to stick to a fixed intensity of 130g/kWh. A discussion of the realistic limits to lithium intensity reduction and the impact of those reductions on the future demand profile would be a useful addition to the analysis.

Reviewer #2 (Remarks to the Author):

While the future energy supply will be based mainly on fluctuating renewable energies, electrical energy storage becomes crucial. In particular, considering future transportation by electrical vehicles, symbiotic coupling of the different energy sectors, and the ongoing electrification of the whole area of energy.

Electrical storage by lithium-ion batteries has shown considerable progress during the last decade. Price reduction of those battery type is unprecedented, approaching now a price level of 10% of the original prices in 2010.

The submitted paper is holistically investigating the resources, limits, and costs of storage by Lithium-ion batteries in a global scale. I consider this paper as rather important, well investigated and recommend it for publication. It could serve policy stakeholders as a base document.

However, some improvements can be made: I pasted all my comments in the PDF submission file which I attached. Furtheron, a reference (development of battery prices) has been missing - I found possible sources, which I attached also. Please consider them for a further version of the paper.

First, we thank the Editor and all Reviewers for their valuable comments.

We have responded to their comments point by point as given below. Blue text gives our response. Red text – new texts added to the original paper. Light blue – texts moved from elsewhere in the document. Single strike – deleted text. Black texts are original text in the paper if they exist in the response.

Any new changes to the Article can also be seen in the Manuscripts version with file name containing “Changesmarked”.

Reviewer #1 (Remarks to the Author):

Overall this is an excellent paper. While it covers previously studied ground using very similar methods it does bring fresh assumptions and contemporary data, giving the analysis merit. The paper is also comprehensive, with broad recognition of the existing literature and sufficient detail regarding assumptions and methods.

Thank you

A few specific points below:

- There is a reference missing in the second paragraph

Thank you. This problem occurs in the PDF version of the document. The reference is clearly shown in our submission as reference 13.

- the second bullet in paragraph 4 page 1 does not make sense to me. Does this mean that previous studies have not done enough to understand other, non-automotive uses of lithium ion batteries in the energy system to meet decarbonisation targets? clarify please

Thank you. Previous studies have not done enough to link Li demand to the latest Li-ion battery driven effort to decarbonize the energy system. The EV penetration as well as the non-automotive demand projections of most of the studies that assess Li demand are also far lower than what is required to achieve meaningful decarbonization by 2050 in addition to missing the latest market trend. Studies partly ignore the ambitious climate targets, and the drastic cost decline of renewable electricity, in particular solar PV, all this leads to stronger policies, stronger economics and a faster transition to (societally) low-cost and sustainable solutions, which requires more short-term battery storage, as projected in most studies. We revised the statement as follow to clarify it.

(ii) ~~weak least~~ attention given to **linking Li demand and the ongoing Li battery driven** the effort to decarbonize the energy sector, among other things;

- The certainty with which the closing statement of paragraph 4 concludes the introduction belies the various uncertainties discussed in the paper, the widely acknowledged challenges of forecasting natural resource production, and the inherent uncertainty in forecasting in general. Particularly given the papers conclusions that most scenarios do not see critical limits to lithium supply in the next two decades. 20 years is sufficient time for a number of changes on the supply or demand side to impact the chance of critical limits to the lithium supply/demand system

Thank you. We made the following modification to the statement:

Finally, it will be shown that Li resource will be a cause of critical limitation **for long-term energy sustainability** without any doubt, if clean energy transition is to be strictly enforced without ameliorating options.

- The ameliorating options mentioned in the paper could be laid out more clearly in the papers conclusions. A bullet list of the things that your analysis suggests would be most useful would be sufficient

Thank you. we have added the following statements to the conclusion.

The mix of these ameliorating measures are: (i) reduce the dependence on LDV and thus LIB by promoting improved public transportation, shared rides and other possible solutions; (ii) recycle once produced LIB by establishing and maintaining efficient recycling systems; (iii) improve LIB technology to reduce material demand per battery capacity; (iv) substitute demand for LIB by developing new battery chemistries; (v) replace demand for batteries by developing sustainable transportation options that do not require batteries.

- It is more accurate to refer to the Cumulative Availability Curve as presenting the available resources given a particularly commodity price, not production cost.

Agreed. Our use of production cost was simply due to lack of data on commodity price at various reserve values. Even though the production cost is also an estimate, that is the easiest way to show that available resource will be driven by price. We have added the following statement to the text to address this gap in the methods section where the subject is discussed.

The available resource, which is represented by the cumulative availability curve, depends on the commodity price of Li. Because of the difficulty to estimate the commodity price, extraction cost is used to show its effect on the cumulative availability curve.

- The paper closes with a statement that lower lithium intensity in batteries could help ease longer term constraints. However, the paper appears to stick to a fixed intensity of 130g/kWh. A discussion of the realistic limits to lithium intensity reduction and the impact of those reductions on the future demand profile would be a useful addition to the analysis.

This was also a point of discussion between authors during this manuscript preparation. The 130 g/kWh value was one of the lowest estimates that our sources show. While several literatures are presenting the expected battery improvement, the level of detail on system level Li demand is not sufficient to make a better guess about future Li intensity reduction. Thus, we couldn't make any further comment on that even though we understand that further reduction is possible. As a result, we decided to refrain from discrediting the analyses by speculating on a too ambitious Li intensity levels without sufficient backing from scientific literatures. At the same time, this is also something that needs a study on its own to establish a proper Li intensity learning curve, as this is a usual practice for silver and silicon demand in solar photovoltaics and other materials in other industries. We have added the following statement to acknowledge the presence of the research gap.

However, future research should employ insights from presently missing Li intensity learning curve when such data are available.

Reviewer #2 (Remarks to the Author):

While the future energy supply will be based mainly on fluctuating renewable energies, electrical energy storage becomes crucial. In particular, considering future transportation by electrical vehicles, symbiotic coupling of the different energy sectors, and the ongoing electrification of the whole area of energy.

Electrical storage by lithium-ion batteries has shown considerable progress during the last decade. Price reduction of those battery type is unprecedented, approaching now a price level of 10% of the original prices in 2010.

The submitted paper is holistically investigating the resources, limits, and costs of storage by Lithium-ion batteries in a global scale. I consider this paper as rather important, well investigated and recommend it for publication. It could serve policy stakeholders as a base document.

Thank you.

However, some improvements can be made: I pasted all my comments in the PDF submission file which I attached. Further on, a reference (development of battery prices) has been missing - I found possible sources, which I attached also. Please consider them for a further version of the paper.

We already have one of your recommended references, e.g. Bloomberg NEF (2019), in our list. It was also correctly cited in our submission but unfortunately the automatised PDF version is partly corrupted. We hope that this won't happen during the revised submission.

Please see our response to all other comments in the corresponding comment window of the PDF file.

Thank you again for your time!